# Fibroblast growth factor receptor expression in hemangioblastomas: A novel therapeutic target

Maya Puttonen[1]*, Olli Tynninen[2], Sami Salmikangas[1], Tiina Vesterinen[2], Harri Sihto[1], Tom Böhling[1]

**1** Department of Pathology, University of Helsinki and Helsinki University Hospital, Helsinki, Finland,
**2** Department of Pathology, HUSLAB, HUS Diagnostic Center, University of Helsinki and Helsinki University Hospital, Helsinki, Finland

* maya.puttonen@helsinki.fi

## Abstract

Hemangioblastoma is a highly vascularized, benign tumor in the central nervous system, frequently associated with von Hippel-Lindau (VHL) disease. Hemangioblastoma may cause tumor-associated hemorrhage or exert pressure on nearby structures, leading to life-threatening complications. Although surgical resection is the primary treatment, complete removal is not always feasible. Accordingly, there is a need to explore targeted or anti-angiogenic therapies. The fibroblast growth factor receptor (FGFR) family has roles in tumorigenesis and angiogenesis, making it a potential target in personalized therapy. The distribution and significance of FGFRs in hemangioblastoma have yet to be investigated. We examined 139 formalin-fixed, paraffin-embedded hemangioblastoma samples from 111 patients, including sporadic cases and those associated with VHL disease. Immunohistochemistry revealed positive staining for FGFR2 (95%) and FGFR4 (61%), while FGFR1 (0%) and FGFR3 (12%) were mainly negative. FGFR2 expression was significantly increased in *VHL*-mutated tumors (75%, $p = 0.034$) and in male patients (68%, $p = 0.020$). Tumors located in the cerebrum ($n = 6$, 5%) had a higher likelihood of positive FGFR4 staining (100%, $p = 0.009$). Additionally, a larger tumor diameter was associated with a higher likelihood of FGFR4 expression (median 12.0 mm vs 17.5 mm, $p = 0.018$), suggesting its contribution in tumor growth. Our study revealed the expression of FGFR2 and FGFR4 in a significant number of hemangioblastomas. This finding demonstrates the potential of FGFRs as promising therapeutic targets for patients with hemangioblastoma.

## Introduction

Hemangioblastoma is a benign, highly vascularized tumor that typically develops in the central nervous system (CNS). The most common location is the cerebellum. Other central nervous system locations include the brainstem, spinal cord, and

**Data availability statement:** All data generated in this study are within the paper and its Supporting Information files. Raw patient data cannot be shared publicly due to privacy protection regulations. These data are available from the Helsinki Biobank (https://helsingin-biopankki.fi/fi/etusivu), with reference to project number HBP20190073, for researchers who meet the criteria for accessing confidential data.

**Funding:** This study was funded by Jane and Aatos Erkko Foundation (https://jaes.fi/en/frontpage/, grant number 4706174, T.B.), Medicinska Understödsföreningen Liv och Hälsa (https://www.livochhalsa.fi/?introduktion, grant number 4708936, T.B.), Finska läkaresällskapet (https://fls.fi/, grant number 4709232, T.B.), Cancer Foundation Finland (https://syopasaatio.fi/en/homepage/for-researchers/, grant number 4709194, H.S.) and Emil Aaltonen Foundation (https://emilaaltonen.fi/, grant number 210179, M.P.). The funders had no role in study design, data collection and analysis, decision to publish, or preparation of the manuscript.

**Competing interests:** The authors have declared that no competing interests exist.

cerebrum, while peripheral involvement includes the retina and nerves [1]. Although commonly sporadic, 20–43% of hemangioblastomas occur in association with von Hippel-Lindau disease (VHL) [2]. VHL is an autosomal dominantly inherited tumor syndrome, usually linked to heterozygosity for a variant in the tumor suppressor gene *VHL* located on chromosome 3p [3,4]. Affected individuals are prone to develop multiple neoplasms, such as CNS hemangioblastomas, renal-cell carcinomas, pheochromocytomas, and neuroendocrine tumors of the pancreas [4]. Despite its benign nature, hemangioblastoma has the potential to induce tumor-associated hemorrhage or compress neighboring structures, thereby posing a risk of fatality [5]. Surgical resection remains the primary treatment option. However, complete removal is not always feasible depending on tumor location [6]. Recurrence of the tumor after resection is more common in VHL patients but can also occur in sporadic cases, emphasizing the need for alternative therapeutic strategies [7]. Among sporadic CNS hemangioblastomas, 4–14% harbor detectable germline *VHL* mutations, and approximately 50% have somatic *VHL* mutations [8–11].

Among the many molecular players involved in tumor development, the fibroblast growth factor receptor (FGFR) family has gained prominence as a target in personalized cancer therapy [12–14]. During the process of carcinogenesis, genetic variations play a role in the upregulation of FGFR mRNA transcription and contribute to the activation of FGFR proteins [12], which holds significance in both tumorigenesis and angiogenesis [15].

The VHL protein normally binds to the hypoxia-inducing transcription factors HIF-1 and HIF-2, marking them for ubiquitination and proteosomal degradation [16]. Dysregulation of VHL function leads to accumulation of HIFs and subsequent overexpression of FGFRs [17] and a variety of growth factors, including platelet-derived growth factor (PDGF) [18,19], vascular endothelial growth factor (VEGF) [20], and erythropoietin (EPO) [21,22]. Upregulation of these factors may lead to angiogenesis and tumorigenesis. Indeed, hemangioblastoma comprises VEGF-expressing stromal cells, and the endothelial cells of the surrounding capillary network express VEGF receptor [23]. FGFRs are also recognized for their involvement in promoting tumor angiogenesis independent of VEGF. For example, FGFRs can act as a compensatory mechanism used by tumors to elude VEGF-targeted therapies [24]. FGF levels in plasma increase prior to disease progression in patients receiving anti-VEGF therapy, indicating a shift in angiogenic dependence from VEGF to FGF signaling [17]. Additionally, FGFRs interact with other cell-surface receptors, including G-protein-coupled receptors and receptor tyrosine kinases, potentially explaining the HIF-VEGF-independent regulation of angiogenesis, resistance to therapy, and metastatic potential of cancer cells [25]. In a mouse model of pancreatic cancer, FGFR signaling bypasses VEGF signaling inhibition, enabling angiogenesis and demonstrating that FGF signaling alone can sustain vascular growth in tumors [26]. Interestingly, Champion et al. reported that *VHL* knockdown in primary human microvascular endothelial cells resulted in defective endocytosis of activated FGFR2, leading to increased cell motility in response to FGF and angiogenic activity. They also knocked down HIF-α in *VHL* loss-of-function endothelial cells, which did

not impede angiogenic activity [27]. *VHL* knockdown in renal cell carcinoma cells resulted in defective internalization and abnormal activation of FGFR1 [28].

A limited number of studies has explored expression levels of FGFRs and their suitability as a treatment target in hemangioblastomas. One study examined a cohort of 20 VHL patients using laser-scanning cytometry and revealed elevated FGFR2 and FGFR3 expression in hemangioblastomas compared with clear-cell renal-cell carcinomas [29]. Another clinical trial involving 6 VHL patients with hemangioblastomas investigated dovitinib, a tyrosine kinase inhibitor that targets FGFR, vascular endothelial growth factor receptor (VEGFR), and platelet-derived growth factor receptor (PDGFR). The trial was terminated due to severe adverse effects, although all 6 patients achieved stable disease [30]. To the best of our knowledge, no other published studies have investigated FGFR expression in hemangioblastomas. Futibatinib and infigratinib are FGFR inhibitors, which have shown efficacy in cholangiocarcinoma with FGFR genetic aberrations [13,31]. These inhibitors could be used for hemangioblastomas, provided further research clarifies the role of FGFR in these tumors.

In this study, formalin-fixed, paraffin-embedded (FFPE) samples obtained from both VHL-related and sporadic hemangioblastoma patients were utilized for immunohistochemical characterization of FGFR1–4. Additionally, the mutation status of the VHL gene was characterized using Sanger sequencing. Corresponding patient data were collected and their association with FGFR1–4 expression patterns were analyzed. Our objective was to contribute to the limited understanding of this area and establish a foundation for further research, facilitating the application of precision medicine to hemangioblastoma.

## Materials and methods

### Ethics approval

This study was approved by the Institutional Ethics Committee of Helsinki University Hospital [HUS/1258/2020] and has therefore been performed in accordance with the Declaration of Helsinki. The collection of materials and data was conducted under an agreement with Helsinki biobank [project number: HBP20190073]. Cause-of-death data were obtained with the approval of the Finnish Social and Health Data Permit Authority Findata [THL/4427/14.02.00/2020]. Informed, project-specific consent was waived since the Finnish Biobank Act provides a lawful basis for research use of biobanked samples.

### Patient samples

With the approval of the Institutional Ethics Committee of Helsinki University Hospital [HUS/430/2021] and under an agreement with Helsinki biobank [project number: HBP20190073] for the transfer of samples and data, all 186 samples from 132 patients and their clinical data were collected for this study. The tumors were diagnosed between 1 January 1983 and 31 December 2018. The median follow-up time was 13.5 years (range 0–39). After excluding tumors that lacked FFPE materials ($n=9$), frozen-section samples ($n=18$), and FFPE samples with insufficient tumor tissue remaining after diagnostics ($n=14$), a total of 145 FFPE hemangioblastoma samples from 132 patients were included in the study (Fig 1). The diagnoses of hemangioblastoma were reviewed by an experienced pathologist (O.T.) and M.P. All methods were performed in accordance with the regulations of Helsinki biobank. Clinical VHL status was considered positive if a patient met clinical diagnostic criteria described elsewhere [32]. Patients with at least one mutation in the *VHL* gene were categorized as *VHL*-mutation positive.

### Immunohistochemistry and scoring

FFPE samples were cut into 4-µm sections and placed on slides. Deparaffinization using xylene, ethanol dehydration in graded concentrations, and incubation in 3% hydrogen peroxide for 30 min were performed. Heat-induced epitope retrieval was conducted using sodium citrate at 95°C for 15 min. The slides were initially incubated overnight at 4°C with primary antibodies diluted in Draco Antibody Diluent (AD500, WellMed, Duiven, the Netherlands). The primary antibodies used in

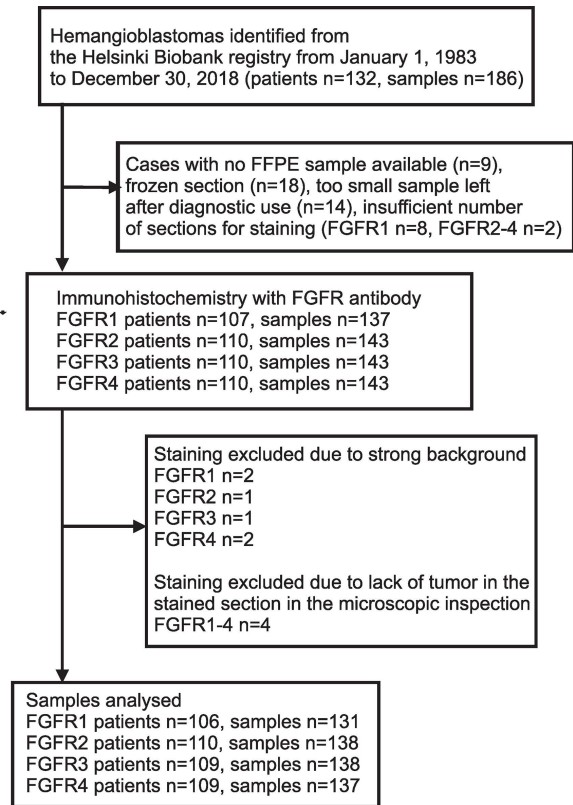

**Fig 1. Flowchart for sample selection.**

this study were FGFR1 mouse monoclonal IgG2a (M2F12, Santa Cruz Biotechnology, Dallas, TX, USA, at a dilution of 1:50), mouse anti-FGFR2 antibody (1G3, Abcam, Cambridge, UK, 1:300), FGFR3 rabbit monoclonal antibody (C51F2, Cell Signaling Technology, Danvers, MA, USA, 1:100), and FGFR4 mouse monoclonal antibody (A-10, Santa Cruz Biotechnology, 1:100). On the following day, the slides were incubated with secondary antibodies for 60 min. For FGFR3 staining, the Orion detection system (rabbit horseradish peroxidase, lot 050619, WellMed) was used. BrightVision poly HRP-Anti-Mouse IgG (lot 230119, WellMed) was used for the other markers. For FGFR1 and FGFR2 staining, an osteosarcoma sample with known overexpression of FGFR1 and FGFR2 served as a positive control. Normal skin tissue and gallbladder tissue were used as positive controls for FGFR3 and FGFR4 staining, respectively. Marker expression was detected by incubating the slides in a DAB Peroxidase Substrate Kit (SK-4105, Vector Laboratories, Newark, CA, USA) for 5 min at room temperature. Hematoxylin was used for counterstaining. Due to insufficient sections, 8 tumors could not be stained with FGFR1 and 2 tumors with FGFR2, FGFR3, and FGFR4 (Fig 1). The stained samples were then examined under a microscope and cytoplasmic staining was scored based on staining intensity using a four-tiered system (score 0 was assigned to negative staining, score 1 to weak staining, score 2 to moderate staining, score 3 to strong staining). Positively stained samples showed a variable combination of staining intensities with intratumoral heterogeneity. The highest score visible in at least 10% of the tumor area in the section was given to the sample. In the analyses using patient-level data, the highest FGFR score among the parallel samples obtained from each patient was used. The positive controls exhibited a cytoplasmic staining pattern as follows: in FGFR1 and FGFR2 stainings, tumor cells, the smooth muscle around blood vessels, and some stromal cells were positive; in FGFR3 staining, the epidermal cells were positive; and in FGFR4 staining, the gallbladder mucosa and blood vessels showed strong positivity, while the muscle layer showed mild positivity.

Due to resource constraints, each staining was done only once. The scoring algorithm was decided by consensus between a senior neuropathologist (OT) and the primary author (MP). Reference slides for each staining category were selected by OT and the remaining samples were scored by MP. Difficult cases were assessed by both MP and OT.

Two samples were excluded from FGFR1 scoring due to occasional strong staining background, along with one sample from FGFR2, one sample from FGFR3, and two samples from FGFR4.

## PCR and Sanger sequencing

Three to four core punches were taken from the tumor area of 105 FFPE blocks from 85 patients. Punches were pre-treated with QS GeneRead DNA FFPE Treatment kit (QIAGEN, Hilden, Germany) and DNA was extracted from the pre-treated punches using QIA Symphony DSP DNA Kit (QIAGEN) according to QIA Symphony LC200 protocol and eluted into 100 µL of TE buffer. PCR for three *VHL* exons was performed with FastStart Taq DNA Polymerase dNTPack kit (Cat#: 04738381001, Roche Diagnostics, Basel, Switzerland). About 20–200 ng of DNA per sample was amplified in a 20-µL reaction in a 96-well format using a PTC-100 thermal cycler (MJ Research, Watertown, MA, USA). The PCR mixture contained 1 × PCR buffer, 20 mM MgCl2, 300 nM forward and reverse primers, 0.2 mM dNTP solution, 1 U of DNA polymerase, and 1 × GC-rich solution. Primer sequences and annealing temperatures are provided in S1 Table. *VHL* exon 1 was sequenced in three parts (1a, 1b, 1c) due to its length and high GC concentration. The PCR cycling conditions were as follows: (1) Initial denaturation at 95°C for 4 min, (2) denaturation at 95°C for 30 s, (3) annealing at 56–57.7 °C for 30 s, (4) elongation at 72°C for 30 s, (5) repeat steps 2–4 for 50 cycles, and (6) final elongation at 72°C for 7 min.

PCR reactions were purified with Applied Biosystems ExoSap-IT PCR Product Cleanup Reagent (Cat#: 78201.1.ML, Thermo Fisher Scientific, Waltham, MA, USA) according to the manufacturer's instructions. Amplified DNA samples were sequenced by Sanger sequencing at the Institute for Molecular Medicine Finland (FIMM, Helsinki, Finland), using an ABI3730xl DNA Analyzer (Thermo Fisher Scientific). The chromatograms were analyzed with Unipro Ugene software version 41.0 [33].

## Statistical analyses

Statistical analyses were conducted to examine the relationships between tumor location and FGFR scores in all samples. The associations between FGFR scores and other factors, including age at first hemangioblastoma diagnosis, sex, clinical VHL status, *VHL* mutation status, and the greatest diameter of all available samples from a patient, were assessed using patient-level data.

A Mann-Whitney $U$ test or a Kruskal-Wallis test was performed as appropriate to evaluate the association between the non-continuous parameters (such as FGFR scores, sex, tumor location, clinical VHL, and VHL mutation status) and the continuous parameters (age and tumor diameter). Effect size r was provided. The association between the non-continuous parameters was evaluated using the Pearson $\chi^2$ test, Fisher's exact test, or Fisher-Freeman-Halton exact test as appropriate. Phi coefficient or Cramer's V was provided as appropriate. Post-hoc testing was performed using a $Z$-test with Bonferroni correction. A Pearson correlation coefficient was calculated to evaluate the relationship between continuous parameters. Survival analysis was performed using the Kaplan-Meier method. Survival data were compared between groups using log-rank tests. All statistical analyses were performed using IBM SPSS Statistics for Windows, version 29 (Chicago, IL, USA). A significance level of $p < 0.05$ was considered statistically significant.

## Results

### Patient and sample characteristics

Two samples were excluded from FGFR1 scoring due to occasional strong staining background, along with one sample from FGFR2, one sample from FGFR3, and two samples from FGFR4 (Fig 1). Additionally, four samples did not exhibit

any visible tumor in the sections under the microscope and were also excluded from further analysis. After these exclusions, the FGFR1 scores were assigned to 131 samples from 106 patients, FGFR2–138 samples from 110 patients, FGFR3–138 samples from 109 patients, and FGFR4–137 samples from 109 patients. A summary of general patient characteristics and individual sample data is provided in Table 1.

### VHL mutation status

DNA sequences were analyzed for 97 tumor samples from 85 patients. *VHL* mutations were detected in 28 patients. *VHL* mutations in our cohort and the number of previously reported mutations in the Cosmic database (accessed on 16 May 2023) are listed in S2 Table.

### FGFR expression

Representative scoring examples and score distributions are shown in Fig 2. None of the tumor tissues stained positive for FGFR1, whereas FGFR1 was consistently expressed in the smooth muscle surrounding blood vessels, Purkinje cells, and the cerebellar cortex (S1 Fig). FGFR2 was positive in 131/138 samples (95%), with scores distributed fairly evenly from 1 to 3. In contrast, FGFR3 was positive in 16/138 samples (12%). Slightly over half (61%) of the samples displayed positive staining for FGFR4. The staining pattern was cytoplasmic for all markers. FGFR3 was positive in the molecular layer of the cerebellar cortex and tumor cells, while FGFR2 was positive in tumor cells, glial cells, and blood vessels, and FGFR4 was positive in tumor cells, astrocytes, and blood vessels. The results revealed no significant associations between FGFR2 and FGFR3 or FGFR4, FGFR3, and FGFR4 (all $p > 0.05$, S3 Table).

### Relationship of patient characteristics with FGFR expression

Results of the statistical analyses are summarized in Table 2. For the statistical analyses, samples were categorized into the following groups: FGFR2 low (score 0 or 1) or high (score 2 or 3), FGFR3 negative (score 0) or positive (scores 1–3), and FGFR4 negative (score 0) or positive (scores 1–3). The deviant categorization was employed for FGFR2 samples due to their even distribution throughout the scores. A significant increase in FGFR2 expression levels was observed in

**Table 1. Patient data.**

| Patients | N = 111 |
|---|---|
| Age at first diagnosis (years) | 48.2 (range 10–83) |
| Sex (M/F) | 71/40 |
| Clinical VHL (pos/neg) | 27/84 |
| *VHL* mutation (pos/neg/no data) | 28/57/26 |
| Tumor greatest diameter (mm) | 17.8 (range 1–50) |
| Survival (alive/deceased) | 34/77 |
| Tumor-specific death | 4 |
| **Samples** | N = 139 |
| Location | |
| cerebellum | 82 |
| spinal cord | 19 |
| cerebrum | 6 |
| brainstem | 15 |
| brain[a] | 16 |
| no data | 1 |

[a]. Specific location data were not available

tumors with a *VHL* mutation ($p=0.034$, Phi coefficient$=0.230$) and in male patients ($p=0.020$, Phi coefficient$=-0.221$). However, no statistically significant associations were found between FGFR2 expression and other factors. None of the factors showed a statistically significant association with FGFR3 expression. Male patients were more likely to exhibit FGFR4 expression ($p=0.02$, Phi coefficient$=-0.222$). Additionally, a larger tumor diameter was associated with a higher likelihood of FGFR4 expression ($p=0.018$, effect size r$=0.247$). Clinical VHL status, *VHL* mutation status, and age at first diagnosis were not associated with FGFR4 expression.

### Relationship of tumor location to FGFR expression

The associations between tumor location and FGFR scores are presented in Table 3. Seventeen samples were excluded from analysis due to lack of specific location data. Tumors of the cerebrum showed a higher likelihood of positive staining for FGFR4 ($p=0.009$, Cramer's V$=0.308$). However, there was no significant association observed between FGFR2 or FGFR3 expression and tumor location. The results revealed no statistically significant associations between sex and tumor location, age and tumor location, sex and age, or the greatest diameter of the tumor and age (all $p>0.05$).

### Association between clinical VHL status, *VHL* mutation, and patient characteristics

Patients with clinical VHL had a significantly younger age at initial hemangioblastoma diagnosis ($p=0.012$, effect size r$=0.310$; S4 Table). There was a positive correlation between *VHL* mutation status and clinical VHL status ($p=0.036$, Phi

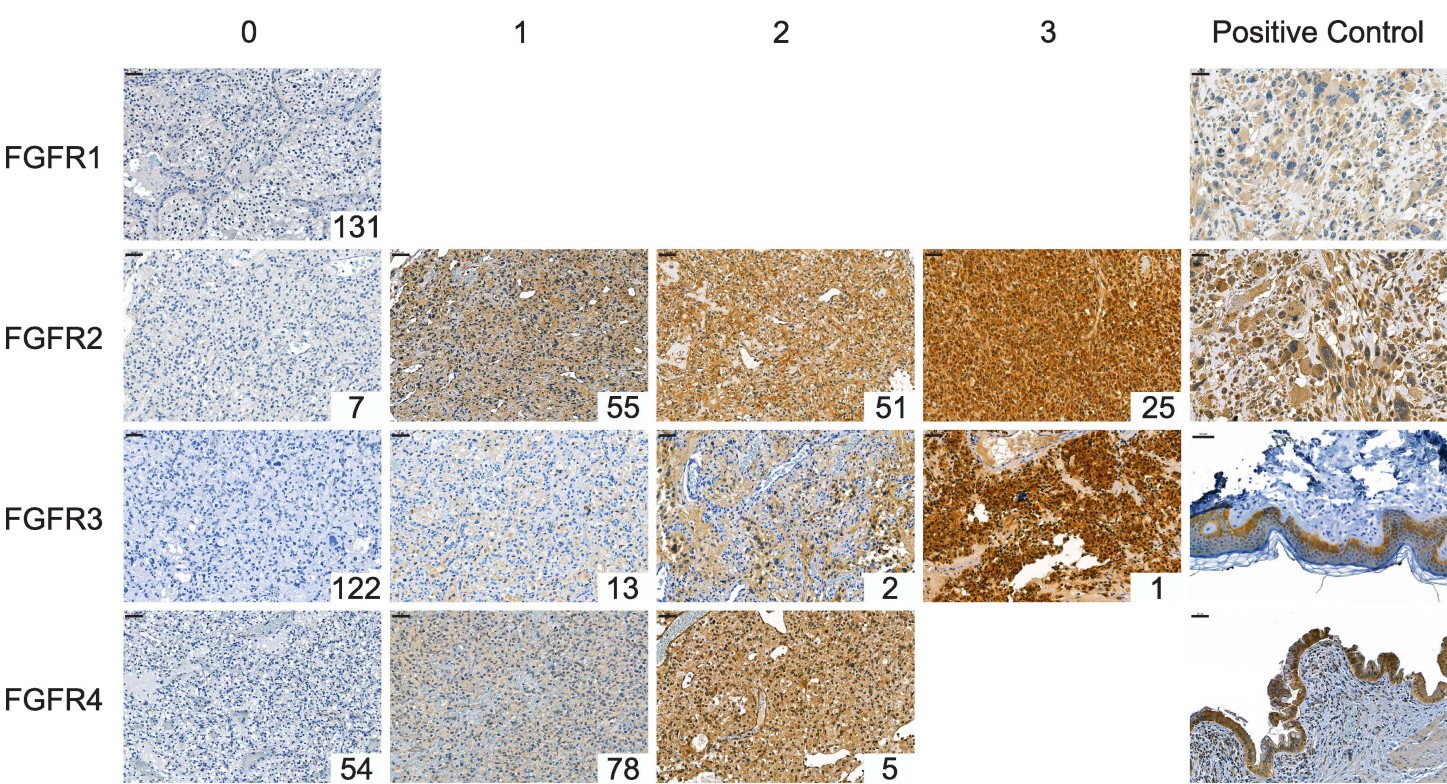

**Fig 2. FGFR immunostainings and scoring.** Representative images of immunohistochemistry are presented, with each column corresponding to a specific score and each row representing a distinct marker. Cytoplasmic staining was evaluated using the following four-tier scale: 0 (negative), 1 (weak), 2 (moderate), and 3 (strong). The number of samples in each scoring group is indicated in the lower-right corner of each image. Positive control tissues are displayed in the rightmost column. Scale bar: 50 μm.

Table 2. Association between patient characteristics with FGFR expression.

| | FGFR2 low^a (n=45)(%) | FGFR2 high^b (n=66)(%) | Total | p | Phi coefficient | FGFR3 neg (n=96)(%) | FGFR3 pos (n=14)(%) | Total | p | Phi coefficient | FGFR4 neg (n=37)(%) | FGFR4 pos (n=73)(%) | Total | p | Phi coefficient |
|---|---|---|---|---|---|---|---|---|---|---|---|---|---|---|---|
| **Sex** | | | | | | | | | | | | | | | |
| female | 22 (55) | 18 (45) | 40 | 0.020c | -0.221 | 34 (87) | 5 (13) | 39 | 1.000d | 0.002 | 19 (48) | 21 (53) | 40 | 0.020c | -0.222 |
| male | 23 (32) | 48 (68) | 71 | | | 62 (87) | 9 (13) | 71 | | | 18 (26) | 52 (74) | 70 | | |
| **Clinical VHL** | | | | | | | | | | | | | | | |
| pos | 7 (26) | 20 (74) | 27 | 0.075c | 0.169 | 23 (85.2) | 4 (15) | 27 | 0.743d | 0.036 | 9 (33) | 18 (67) | 27 | 0.969c | 0.004 |
| neg | 38 (45) | 46 (55) | 84 | | | 73 (88.0) | 10 (12) | 83 | | | 28 (34) | 55 (66) | 83 | | |
| **VHL mutation** | | | | | | | | | | | | | | | |
| pos | 7 (25) | 21 (75) | 28 | 0.034c | 0.230 | 23 (82.1) | 5 (18) | 28 | 0.519d | 0.075 | 9 (32) | 19 (68) | 28 | 0.867c | -0.180 |
| neg | 28 (49) | 29 (51) | 57 | | | 50 (87.7) | 7 (12) | 57 | | | 17 (30) | 39 (70) | 56 | | |
| | low mean rank (range) | high mean rank (range) | | p | Effect size r | neg mean rank (range) | pos mean rank (range) | | p | Effect size r | neg mean rank (range) | pos mean rank (range) | | p | Effect size r |
| Age at first diagnosis | 60.81 (10-77) | 52.72 (16-83) | | 0.193e | 0.123 | 54.13 (10-83) | 64.89 (26-71) | | 0.238e | 0.113 | 50.81 (10-75) | 57.88 (16-83) | | 0.272e | 0.105 |
| Greatest diameter (mm) | 52.73 (4.5-50) | 42.31 (1-40) | | 0.066e | 0.192 | 44.52 (1-50) | 54.88 (10-35) | | 0.189e | 0.138 | 36.94 (1.5-50) | 50.68 (1-40) | | 0.018e | 0.247 |
| Not available | 8 | 11 | | | | 18 | 1 | | | | 6 | 13 | | | |

a. Scores 0 or 1
b. Scores 2 or 3
c. Chi-square
d. Fisher's exact test
e. Mann-Whitney U

**Table 3. The associations between tumor location and FGFR scores.**

| | FGFR2 | | | | | FGFR3 | | | | | FGFR4 | | | | |
|---|---|---|---|---|---|---|---|---|---|---|---|---|---|---|---|
| | low[a] (n=62) (%) | high[b] (n=76) (%) | total | p | Cramer's V | neg (n=122) (%) | pos (n=16) (%) | total | p | Cramer's V | neg (n=55) (%) | pos (n=82) (%) | total | p | Cramer's V |
| Location | | | | | | | | | | | | | | | |
| cerebellum | 41 (50) | 41 (50) | 82 | 0.389[c] | 0.163 | 69 (85) | 12 (15) | 81 | 0.261[c] | 0.196 | 26 (32) | 55 (68) | 81 | 0.009[c] | 0.308 |
| spinal cord | 7 (37) | 12 (63) | 19 | | | 19 (100) | 0 (0) | 19 | | | 12 (63) | 7 (37) | 19 | | |
| cerebrum | 1 (17) | 5 (83) | 6 | | | 6 (100) | 0 (0) | 6 | | | 0 (0) | 6 (100) | 6 | | |
| brain stem | 7 (47) | 8 (53) | 15 | | | 14 (93) | 1 (7) | 15 | | | 8 (53) | 7 (47) | 15 | | |

a. Scores 0 or 1

b. Scores 2 or 3

c. Fisher-Freeman -Halton Exact test

coefficient = 0.228). Of the 61 individuals who did not meet the clinical VHL criteria, 16 patients (26%) had a *VHL*-mutated tumor. Conversely, among the 24 individuals who met the clinical VHL criteria, 12 patients (50%) harbored a *VHL* mutation in the tumor. Notably, patients without clinical VHL were significantly more likely to develop hemangioblastoma in the cerebellum and the spinal cord than those who met the clinical VHL criteria ($p = 0.008$, Cramer's V = 0.310). There was no statistically significant association between *VHL* mutation status and age, sex and clinical VHL, sex and *VHL* mutation status, or tumor location and *VHL* mutation status (all $p > 0.05$, S4 Table).

## Survival analysis

The median survival time was 13.5 years (range 0–34.1); 19 patients passed away during the follow-up period. Disease-specific fatality was detected in 3 patients. The documented causes of death were cerebellar hemorrhage, brain compression, and infratentorial benign neoplasms. None of the FGFR expression data or other factors showed any statistically significant association with overall patient survival (S2 Fig). The association between disease-specific survival and FGFR expression was not tested due to the small number of patients with disease-specific fatality.

## Discussion

In this study, we investigated the associations between FGFR immunostaining and various patient and sample characteristics in hemangioblastoma, providing novel insights into FGFR expression in this context. Our analysis revealed FGFR2 expression in most of the hemangioblastoma samples, while FGFR3 were mostly negative. FGFR1 was negative in all samples. Approximately half of the samples had positive staining for FGFR4. The staining patterns demonstrated intratumoral heterogeneity, consistent with observations in other tumor types [34,35].

FGFR signaling plays a pivotal role in cancer biology by activating key pathways such as JAK/STAT, PI3K/AKT/mTOR, and RAS-RAF-MEK-MAPK. These pathways regulate essential processes, including cell proliferation, survival, metabolism, angiogenesis, and epithelial-mesenchymal transition [36,37]. Dysregulation of FGFR signaling contributes to tumor progression and resistance to therapy through mechanisms including suppression of apoptosis, increased drug efflux, and alterations in cell-cycle regulation [38,39].

Aberrations in FGFR2 and FGFR4 are implicated in various tumors, including those of the CNS. In gliomas, FGFR2 downregulation is associated with increased proliferation and reduced survival [40]. FGFR4 expression is physiologically low in the brain but is increased in tumors such as glioblastoma, where it enhances integrin-mediated cell adhesion and

invasion [41,42]. Furthermore, cerebral tumors may rely on FGFR4-driven angiogenesis to promote blood vessel formation [43,44]. Aberrant FGFR4 activity drives activation of signaling pathways that promote cell proliferation, survival, and invasion, as observed in epithelial cancers such as non-small cell lung cancer, breast cancer, and prostate cancer [45,46]. Notably, in this study, cerebral hemangioblastomas had a higher likelihood of positive staining for FGFR4, suggesting that the expression characteristics of FGFR may vary depending on tumor location. Regional differences in blood-brain barrier (BBB) properties may influence how tumors affect BBB integrity and permeability, potentially leading to varying FGFR levels across brain regions [47].

Although the complete absence of FGFR1 was unexpected, various aberrations may affect protein expression levels or its recognition by antibodies. A unique FGFR1 alteration involving tail-to-tail rearrangements that delete the ectodomain was identified in squamous cell lung cancer. The resulting truncated FGFR1 protein retains oncogenic activity, driving tumor growth in experimental models. Importantly, tumors harboring these alterations demonstrated high sensitivity to FGFR inhibitors, suggesting potential for targeted therapy and a biomarker for personalized treatment [48]. Mutations resulting in FGFR1 loss were observed in a subset of hemangioblastoma samples analyzed using single-nucleotide polymorphism microarrays and droplet digital PCR [49]. Similarly, microdeletions of FGFR1 have been identified in myeloid and lymphoid neoplasms [50]. In astrocytic tumors, FGFR1 expression increases with malignancy, with lower-grade tumors showing weaker expression [51,52]. FGFR1 is a key driver in aggressive ependymomas [53]. The association of FGFR1 with aggressive behavior aligns with its absence in histologically benign hemangioblastomas, although differences in histogenesis complicate direct comparisons.

Significant advancements have been made in the development of drugs targeting FGFR inhibition. For instance, in patients diagnosed with *FGFR2* fusion- or rearrangement-positive cholangiocarcinoma, significant improvements in patient outcomes were observed in treatment with futibatinib, a covalently binding inhibitor of FGFRs [54], and infigratinib, a selective, ATP-competitive inhibitor of FGFRs [55]. Pemigatinib, a small molecule inhibitor of FGFR1–3, has been approved in the US for treatment of unresectable cholangiocarcinoma with *FGFR2* alterations, while erdafitinib, pan-FGFR inhibitor, is indicated for metastatic urothelial carcinoma with *FGFR2* and *FGFR3* alterations [56,57]. Encouraging results from trials have been reported for FGFR4-specific inhibitors in patients with hepatocellular carcinoma [58–60]. Although the initial trial with dovitinib, a pan-tyrosine kinase inhibitor targeting FGFR, VEGFR, and other receptor tyrosine kinases, was discontinued due to serious side effects [30], our findings strongly support the continued exploration of FGFR inhibitors as therapeutic options for hemangioblastoma patients.

We observed that tumors with *VHL* mutations exhibited a higher likelihood of increased FGFR2 expression. A previous study reported that knockdown of *VHL* in primary human microvascular endothelial cells led to a 3-fold increase of surface FGFR2, leading to increased angiogenic activity in response to fibroblasts or through increased ERK1/2 signaling and ETS1 activity [27]. FGFR2 can suppress cancer cell migration by modulating HIF signaling. However, *VHL* knockdown in endothelial cells disrupts FGFR2 endocytosis, resulting in elevated cell motility and increased angiogenic activity. Together with these previous reports, our results suggest a potential role of FGFR2 in the development and progression of *VHL*-mutated hemangioblastomas. Furthermore, a positive correlation was identified between tumor size and FGFR4 expression, suggesting a possible role of FGFR4 in tumor progression in hemangioblastoma. Supportive evidence for the role of FGFR4 has been shown in studies of other tumors. For instance, FGFR4-positive staining is associated with postoperative residual disease in ovarian cancer [61]. In pituitary adenomas, a significant correlation was observed between high levels of FGFR4 expression and the proliferation marker Ki-67, and FGFR4 expression is more prevalent in invasive tumors [62].

Interestingly, in our cohort, male patients were more likely to exhibit FGFR2 and FGFR4 expression. FGFRs are involved in sex determination and the development of sex-specific organs, possibly contributing to the observed differences in expression levels. FGFR2 is involved in male sex determination by acting as the receptor for FGF9 [63]. FGFRs are essential in sperm development and maturation, epididymal function, and prostate development [64]. FGFR signaling

is also involved in steroid hormone-dependent development of mammary ducts [65]. Sex-specific differences in the genetic profile of FGF receptors have been reported in tumors of the liver, lung, urinary bladder, and larynx [66–68]. For instance, FGFR and its related pathways are amplified in hepatocellular carcinomas in males [69]. Our results suggest the possibility of similar sex-related differences in the genetics of FGFR in hemangioblastoma. Further genetic studies on the subject are warranted.

Survival analysis demonstrated that FGFR expression did not show a statistically significant association with overall patient survival. It is important to note that these analyses were limited by the relatively small cohort size, the low number of deaths, and the benign nature of hemangioblastoma, where mortality is typically caused by events such as intracranial bleeding or brain herniation rather than the tumor itself [70–72]. Although the tumors are benign, treatment success and prognosis are poorer for tumors in surgically challenging locations, such as the spinal nerves and brain stem [73,74]. Additionally, incomplete resections are associated with a higher level of postoperative bleeding, tumor recurrence, and other adverse outcomes [75]. Tumors associated with VHL syndrome also have a higher occurrence of unfavorable outcomes [76].

As expected, we found a positive correlation between the presence of somatic *VHL* mutations and clinical VHL status. Patients with clinical VHL developed hemangioblastoma at a younger age, supporting the fact that VHL-associated tumors occur at a younger age than sporadic hemangioblastomas [77]. In our series, 50% of patients who met the clinical VHL criteria harbored a *VHL* mutation, while 26% of patients who did not meet the criteria had a *VHL* mutation. This finding aligns with a previous study utilizing Sanger sequencing, which reported a mutation frequency of 64% in VHL-related hemangioblastomas and 19% in sporadic hemangioblastomas [78]. In the previous study, the authors performed targeted deep sequencing and multiplex ligation-dependent probe amplification in addition to Sanger sequencing. They identified VHL mutations in 100% of VHL cases and in 62% of sporadic cases [78], suggesting that some mutations cannot be identified by Sanger sequencing alone. This limitation could weaken the reliability of our analyses, particularly the association between *VHL* mutation status and FGFR expression.

Another limitation of our study is its retrospective nature, which may have introduced biases inherent to data collection and analysis. Additionally, the visual evaluation of immunohistochemical stains relies on subjective interpretation, which may affect reproducibility. Incorporating modern digital pathology tools may have mitigated this limitation by providing more reproducible, objective, and quantitative data. Furthermore, our survival analysis did not reveal any statistically significant association between FGFR expression or other factors and overall patient survival. This analysis was limited by the relatively small cohort size, the low number of deaths, and the benign nature of hemangioblastoma, where mortality is typically caused by events such as intracranial bleeding or brain herniation rather than the tumor itself [70–72]. Despite these limitations, our findings suggest heterogeneity in the expression profiles of FGFRs in hemangioblastomas. To further explore this hypothesis, transcriptomic studies focusing on FGFR expression and the downstream pathway enrichment in hemangioblastoma are essential. Such investigations could pave the way for novel pharmacological treatments, particularly for surgically challenging tumors, such as those located in the brainstem or spinal cord.

In conclusion, our study highlighted the frequent expression of FGFR2 and FGFR4 in hemangioblastoma and revealed a correlation between larger tumor size and increased FGFR4 expression, suggesting their roles in tumor development and growth. These findings underscore the potential of FGFRs as therapeutic targets in hemangioblastoma. Future research, including comprehensive genomic and transcriptomic analyses and clinical trials, will be crucial to validate these results and further elucidate the biological and therapeutic implications of FGFRs in the management of this rare tumor.

## Supporting information

**S1 Table. Primer sequences and annealing temperatures.**
(PDF)

**S2 Table. Mutation status of *VHL* gene.**
(PDF)

**S3 Table. Confounding factors.**
(PDF)

**S4 Table. Association of VHL status with tumor characteristics.**
(PDF)

**S1 Fig. FGFR1 staining in normal structures. a and b.** Smooth muscle of the blood vessels stained positive. Scale bar = 100 μm. c. The black arrowhead points to a positively stained Purkinje cell. The white arrow indicates an area where the white matter of the cerebellum is positively stained. The black arrow marks a blood vessel with positively stained smooth muscle. Scale bar = 100 μm. d. Purkinje cells were positively stained. Scale bar = 50 μm. e. The white matter of the cerebellum stained positive. Scale bar = 100 μm.
(TIF)

**S2 Fig. Kaplan-Meier survival analyses. a.** Association between FGFR2 expression and overall survival, b. association between FGFR3 expression and overall survival, c. association between FGFR4 expression and overall survival.
(TIF)

## Acknowledgments

DNA extraction and sequencing were performed at the Institute for Molecular Medicine Finland FIMM Genomics unit supported by HiLIFE and Biocenter Finland. Helsinki Biobank is acknowledged for provision of the research materials.

## Author contributions

**Conceptualization:** Maya Puttonen, Harri Sihto.

**Formal analysis:** Maya Puttonen.

**Funding acquisition:** Maya Puttonen, Harri Sihto, Tom Böhling.

**Investigation:** Maya Puttonen, Olli Tynninen, Sami Salmikangas.

**Methodology:** Maya Puttonen, Olli Tynninen, Sami Salmikangas, Harri Sihto.

**Project administration:** Maya Puttonen.

**Resources:** Tiina Vesterinen, Harri Sihto.

**Supervision:** Tiina Vesterinen, Harri Sihto, Tom Böhling.

**Validation:** Olli Tynninen, Tiina Vesterinen, Harri Sihto, Tom Böhling.

**Visualization:** Maya Puttonen.

**Writing – original draft:** Maya Puttonen.

**Writing – review & editing:** Maya Puttonen, Olli Tynninen, Sami Salmikangas, Tiina Vesterinen, Harri Sihto, Tom Böhling.

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
