## [Decision Letter · Decision Letter 0]

1 Nov 2024

PONE-D-24-35229Fibroblast growth factor receptors 2 and 4 are frequently expressed in hemangioblastomaPLOS ONE

Dear Dr. Puttonen,

Thank you for submitting your manuscript to PLOS ONE. After careful consideration, we feel that it has merit but does not fully meet PLOS ONE’s publication criteria as it currently stands. Therefore, we invite you to submit a revised version of the manuscript that addresses the points raised during the review process.

We look forward to receiving your revised manuscript.

Kind regards,

Md Shaifur Rahman, Ph.D

Academic Editor

PLOS ONE

Journal Requirements:

Reviewers' comments:

Reviewer's Responses to Questions

**Comments to the Author**

1. Is the manuscript technically sound, and do the data support the conclusions?

Reviewer #1: Yes

Reviewer #2: Partly

2. Has the statistical analysis been performed appropriately and rigorously? 

Reviewer #1: Yes

Reviewer #2: Yes

3. Have the authors made all data underlying the findings in their manuscript fully available?

Reviewer #1: Yes

Reviewer #2: No

4. Is the manuscript presented in an intelligible fashion and written in standard English?

Reviewer #1: Yes

Reviewer #2: Yes

5. Review Comments to the Author

Reviewer #1: The study by Puttonen et al. investigates the expression of FGF-receptors 1-4 in FFPE tissues of a hemangioblastoma cohort by immunohistochemistry.

The study data are novel and the study appears thoroughly performed and is well described.

Here are my comments:

While staining patterns are more or less in line with what has been reported in other tumor types for FGFR2-4, the complete absence of FGFR1 expression is puzzling, but would be an interesting finding if confirmed. Maybe the authors could assess at least in a few cases whether FGFR1 transcript is also missing or expressed at very low levels compared to the other FGFRs.

While the link between FGFR2 and VHL and the link between FGFR4 and tumor diameter are interesting, functional data would be required for concluding that FGFRs are indeed causally involved in tumor growth in this case. Is it possible to deduce whether FGFR expression is increased or decreased compared to non-tumor counterparts (probably difficult in this tumor type), or more intense or more frequent in more advanced/recurrent tumors?

Reviewer #2: Comments and recommendations:

► The manuscript could benefit from a thorough proofread to correct minor errors in grammar, punctuation, and formatting. For example, there are several instances of missing articles, incorrect verb tenses, and inconsistent formatting of headings and subheadings.

► The study addresses an important gap related to FGFRs in hemangioblastoma. However, the research gap and objectives are not explicitly highlighted in the introduction.

► The introduction could be strengthened by providing a clearer background on the significance of hemangioblastomas and the role of FGFRs in tumor development.

► Expand on the biological significance of FGFRs in tumor growth and angiogenesis resistance to VEGF therapies.

► The FGFR1-negative result should be discussed in relation to other studies. Are there other tumors where FGFR1 is similarly inactive?

► Some sections are slightly verbose, especially in the abstract and discussion. Reducing redundancy will enhance clarity.

► Some of the statistical conclusions require further elaboration to strengthen their relevance.

► The conclusion could be improved by summarizing the main findings and implications of the study. Consider adding a final thought on the significance of the study and its contributions to the field.

Title:

► Comment: The title is clear and concise but could be improved by adding subtitles to provide more context. For example, "Fibroblast Growth Factor Receptors (FGFRs) Expression in Hemangioblastomas: A Novel Therapeutic Target"

Abstract:

Line 1: “Hemangioblastoma is a highly vascularized benign tumor...”

• Comment: The description is concise but needs more specificity. Mention the risks associated with the tumor, like hemorrhage and recurrence, which create a clinical need for alternative treatments.

Line 3: “...complete removal is not always feasible, necessitating alternative strategies.”

• Comment: Avoid vague phrasing like “alternative strategies.”. Specify what strategies you are referring to (e.g., targeted therapies, FGFR inhibition). Instead: “...necessitating exploration of FGFR-targeted therapies or angiogenesis inhibitors.”

Line 9:

"A significant increase in FGFR2 expression level was observed in tumors harboring a VHL mutation..."

• Scientific Suggestion: You could discuss the biological mechanisms through which FGFR2 might interact with VHL mutation pathways. Expanding on the molecular links between the FGFR and hypoxia-inducible factor pathways would deepen the relevance of the finding.

Line 9:

"A significant increase in FGFR2 expression level was observed in tumors harboring a VHL mutation..."

• Comment: Consider rephrasing to avoid redundancy: "FGFR2 expression significantly increased in VHL-mutated tumors."

Line 13:

"This finding highlights the potential of FGFRs as promising therapeutic targets..."

• Comment: Use a more assertive tone. Replace "highlights" with "supports" or "demonstrates" for a stronger conclusion.

• Grammar: “The distribution and significance of FGFRs in hemangioblastoma is yet to be investigated” → Should be “are yet to be investigated” (subject-verb agreement).

• Scientific Improvement: The statistical significance for FGFR2 in tumors with VHL mutations (p=0.034) is mentioned. Consider including confidence intervals to enhance clarity.

Introduction:

Comment: The introduction is not clearly structured and does not provide a clear background on the significance of hemangioblastomas and the role of FGFRs in tumor development. Consider adding more context to the introduction to explain the importance of studying FGFRs in hemangioblastomas.

► Mention drugs like infigratinib or futibatinib to connect findings with clinical therapies and show the relevance of your work.

► Provide a detailed explanation of the biological significance of FGFRs in VHL-mutated hemangioblastomas, which bypass VEGF pathways and explain resistance to anti-VEGF therapies.

► Clarify variability in VHL mutation frequency and cite relevant studies.

Line 19:

"The most common location is the cerebellum, and other locations include the brainstem, spinal cord, cerebrum, retina, and peripheral nerves."

• Comment: Consider separating peripheral and central nervous system structures for clarity: "Other central nervous system locations include the brainstem, spinal cord, and cerebrum, while peripheral involvement includes the retina and nerves."

Line 30:

"...has gained prominence as a target in personalized cancer therapy."

• Comment: Mention specific FGFR inhibitors (e.g., infigratinib, futibatinib) used in clinical trials for other cancers. This shows the translational potential of FGFR inhibition and can be linked to future therapeutic options for hemangioblastomas.

Line 30:

"...has gained prominence as a target in personalized cancer therapy."

• Comment: This is a strong claim. Consider adding more references to support the statement.

Line 45:

"They also knocked down HIF-α in VHL loss-of-function endothelial cells, which did not impede their angiogenic activity."

• Scientific Comment: This is a crucial point but needs further explanation. Why does HIF-independent angiogenesis driven by FGFR matter in the context of VHL-mutated hemangioblastomas? Consider elaborating on why this could explain resistance to VEGF-targeted therapies.

Methods:

Line 90: “FFPE samples were cut into 4 µm sections...”

• Scientific Comment: Specify how variability in sample preparation (e.g., staining intensity) was managed to ensure reproducibility. Was scoring done by multiple pathologists?

Line 153:

"Two samples were excluded from FGFR1 scoring due to occasional strong staining background..."

• Comment: Background staining can be a technical issue. Was there an attempt to optimize antibody dilution or perform antigen retrieval differently to improve FGFR1 staining? Addressing this could suggest better reproducibility.

Line 180:

"...samples were categorized into the following groups: FGFR2 low (score 0 or 1) or high (score 2 or 3)."

• Comment: It's important to justify these cutoffs based on existing literature or statistical analysis. Were these categories pre-defined based on prior studies, or was there an exploratory analysis? Justifying this strengthens the validity of the results.

Line 188:

"Male patients were more likely to exhibit FGFR4 expression (p = 0.02)."

• Comment: Was the impact of hormonal differences between male and female patients explored? FGFR expression can be regulated by hormone levels, so this could be an interesting angle to pursue in the discussion or future studies.

Survival Analysis:

• While mortality is rare in hemangioblastoma, survival analyses could be enhanced by exploring predictors like tumor location or resection completeness.

Results

Line 193:

"Tumors of the cerebrum showed a higher likelihood of positive staining for FGFR4 (p = 0.009)."

• Comment: Consider providing effect sizes or odds ratios alongside p-values for more informative statistical reporting.

Line 193:

"Tumors of the cerebrum showed a higher likelihood of positive staining for FGFR4 (p = 0.009)."

• Comment: Explore why cerebrum-localized tumors might have higher FGFR4 expression. Could it be linked to tumor microenvironment differences, such as blood-brain barrier permeability or vascular density?

Line 223:

"Approximately half of the samples displayed positive staining for FGFR4."

• Comment: This could be discussed in the context of FGFR4-related tumor biology in other cancers. Is there a known role for FGFR4 in tumor progression or angiogenesis that could support this finding? Connecting this to broader cancer biology would elevate the discussion.

Discussion:

Comment: The discussion could be strengthened by providing a more detailed analysis of the results and their implications for the field. Consider discussing the limitations of the study and potential avenues for future research.

► The discussion section lacks specific details, such as the potential mechanisms by which FGFRs may contribute to the development and progression of hemangioblastomas.

► Throughout the results, p-values are provided, but without context for the effect size or clinical significance. Strengthening statistical sections by providing effect sizes or confidence intervals will improve the interpretation.

► While FGFR2 and FGFR4 are reported as overexpressed, there is limited mechanistic discussion. Suggest exploring downstream pathways activated by FGFRs, which could contribute to hemangioblastoma growth or treatment resistance. Expand on FGFR signaling pathways: Integrate more mechanistic insights by discussing the MAPK and PI3K pathways that FGFRs activate and their role in tumor progression.

► Cite any studies that report sex-dependent FGFR expression in other tumors to strengthen your argument.

► Suggest future research on genomic and transcriptomic profiling of FGFRs to explore their therapeutic potential.

► Go beyond descriptive reporting of FGFR2 and FGFR4 expression—discuss the implications for tumor growth, angiogenesis, or potential drug resistance.

► Be more decisive in suggesting FGFR inhibitors for future clinical research, emphasizing the potential impact of your findings.

► Add a section highlighting any limitations (e.g., retrospective design, manual scoring bias) to pre-empt reviewer concerns and demonstrate scientific integrity.

Line 220: “FGFR2 was expressed in the majority of samples, while FGFR1 was negative.”

• Scientific Comment: This result should be discussed in relation to other tumors—does this pattern align with known FGFR expression in other CNS tumors?

Line 240:

"Together with this previous report, our result indicates a potential role of FGFR2 in the development and progression of VHL-mutated hemangioblastomas."

• Comment: It would be useful to mention specific signaling pathways involved (e.g., FGFR2-mediated MAPK or PI3K pathways). This would provide a more detailed mechanistic insight.

Line 250:

"Interestingly, in our cohort, male patients were more likely to exhibit FGFR2 and FGFR4 expression."

• Comment: This observation could be strengthened by citing any preclinical or clinical evidence supporting differential FGFR expression based on sex in other tumors. If no such evidence exists, suggest this as a potential avenue for future research.

Line 260:

"...exhibited a higher likelihood of positive staining for FGFR4, suggesting that the expression characteristics of FGFR may vary depending on tumor location."

• Comment: It may be worth exploring whether tumor location in the brain affects its access to growth factors, leading to differential FGFR expression. Adding speculation here opens up new hypotheses for future studies.

Conclusion

• Comment: Instead of “highlighting” potential targets, use stronger phrasing:

o Example: “Our findings indicate that FGFR2 and FGFR4 are promising therapeutic targets...”

► Suggest conducting clinical trials using FGFR inhibitors to validate their potential in treating hemangioblastomas.

►Add a brief paragraph acknowledging the retrospective nature and the subjectivity of manual scoring.

Formatting and Data Presentation

► Table 3 is somewhat hard to read due to misaligned columns. Clearer formatting will improve readability.

► Ensure all figures mentioned (e.g., Fig. 1 and Fig. 2) are adequately described in the text. Also, the legend for Figure 2 is too brief and lacks sufficient detail to guide readers. Expand the legend by describing each panel thoroughly—what specific staining intensities are shown, what the comparison groups represent, and how scoring was performed. This will ensure clarity and alignment with high technical reporting standards.

This manuscript addresses a relevant topic with significant clinical potential. However, several areas need improvement in language, scientific rigor, and presentation. With these refinements, the study can make a valuable contribution to the field, and it can meet the rigorous standards required for PLOS ONE publication.

6. PLOS authors have the option to publish the peer review history of their article (what does this mean? ). If published, this will include your full peer review and any attached files.

**Do you want your identity to be public for this peer review?** For information about this choice, including consent withdrawal, please see our Privacy Policy .

Reviewer #1: No

Reviewer #2: No

---

## [Author Response · Author response to Decision Letter 1]

6 Feb 2025

Dear Editor,

Thank you very much for the opportunity to revise and resubmit our manuscript, and for the detailed feedback from the reviewers. We have now revised the manuscript ID PONE-D-24-35229 “Fibroblast growth factor receptors 2 and 4 are frequently expressed in hemangioblastoma” by Puttonen, Tynninen, Salmikangas, Vesterinen, Sihto, and Böhling according to reviewers’ suggestions. We sincerely hope that our revised manuscript will be suitable for publication in PLOS ONE.

Changes in text are highlighted in red.

Your faithfully,

Maya Puttonen

Reviewer Comments:

Reviewer #1: The study by Puttonen et al. investigates the expression of FGF-receptors 1-4 in FFPE tissues of a hemangioblastoma cohort by immunohistochemistry.

The study data are novel and the study appears thoroughly performed and is well described.

Here are my comments:

While staining patterns are more or less in line with what has been reported in other tumor types for FGFR2-4, the complete absence of FGFR1 expression is puzzling, but would be an interesting finding if confirmed. Maybe the authors could assess at least in a few cases whether FGFR1 transcript is also missing or expressed at very low levels compared to the other FGFRs.

While the link between FGFR2 and VHL and the link between FGFR4 and tumor diameter are interesting, functional data would be required for concluding that FGFRs are indeed causally involved in tumor growth in this case. Is it possible to deduce whether FGFR expression is increased or decreased compared to non-tumor counterparts (probably difficult in this tumor type), or more intense or more frequent in more advanced/recurrent tumors?

Re: Unfortunately, mRNA from the biobank tissue material was not available to us, preventing the verification of FGFR1 expression levels. Instead, we have included a more detailed description of FGFR1 expression in the tumor-surrounding tissues within the tumor samples, as well as in the positive control in the results (S1 Fig). We also discussed the absence of FGFR1 expression in a new paragraph at the discussion section.

Lines 170–172: “None of the tumor tissues stained positive for FGFR1, whereas FGFR1 was consistently expressed in the smooth muscle surrounding blood vessels, Purkinje cells, and the cerebellar cortex (S1 Fig).”

Lines 241–250: “Although the complete absence of FGFR1 was unexpected, various aberrations may affect protein expression levels or its recognition by antibodies...”

There was no statistically significant difference in FGFR staining intensities between primary and recurrent tumors.

FGFR3 FGFR4 FGFR2

Pos Neg Total p Phic Pos Neg Total p Phic High Low Total p Phic

Tumor type

Primary tumors 15 97 112 0.306a 0.117 71 41 112 0.074b 0.153 61 52 113 0.584b -0.047

Recurrent tumors 1 25 26 11 14 25 15 10 25

Total 16 122 138 82 55 137 16 122 138

a. Fisher's exact test

b. Chi-square

c. Phi coefficient

Reviewer #2: Comments and recommendations:

► The manuscript could benefit from a thorough proofread to correct minor errors in grammar, punctuation, and formatting. For example, there are several instances of missing articles, incorrect verb tenses, and inconsistent formatting of headings and subheadings.

Re: The entire text has been proofread and improved by a professional language editor.

► The study addresses an important gap related to FGFRs in hemangioblastoma. However, the research gap and objectives are not explicitly highlighted in the introduction.

► The introduction could be strengthened by providing a clearer background on the significance of hemangioblastomas and the role of FGFRs in tumor development.

► Expand on the biological significance of FGFRs in tumor growth and angiogenesis resistance to VEGF therapies.

► The FGFR1-negative result should be discussed in relation to other studies. Are there other tumors where FGFR1 is similarly inactive?

► Some sections are slightly verbose, especially in the abstract and discussion. Reducing redundancy will enhance clarity.

► Some of the statistical conclusions require further elaboration to strengthen their relevance.

► The conclusion could be improved by summarizing the main findings and implications of the study. Consider adding a final thought on the significance of the study and its contributions to the field.

Re: We thank the reviewer for valuable comments. Below, we address them one by one in detail.

Title:

► Comment: The title is clear and concise but could be improved by adding subtitles to provide more context. For example, "Fibroblast Growth Factor Receptors (FGFRs) Expression in Hemangioblastomas: A Novel Therapeutic Target"

Re: The title has been improved as suggested.

Abstract:

Line 1: “Hemangioblastoma is a highly vascularized benign tumor...”

• Comment: The description is concise but needs more specificity. Mention the risks associated with the tumor, like hemorrhage and recurrence, which create a clinical need for alternative treatments.

Line 3: “...complete removal is not always feasible, necessitating alternative strategies.”

• Comment: Avoid vague phrasing like “alternative strategies.”. Specify what strategies you are referring to (e.g., targeted therapies, FGFR inhibition). Instead: “...necessitating exploration of FGFR-targeted therapies or angiogenesis inhibitors.”

Re: We have revised the abstract to address the reviewers' comments.

Line 9:

"A significant increase in FGFR2 expression level was observed in tumors harboring a VHL mutation..."

• Scientific Suggestion: You could discuss the biological mechanisms through which FGFR2 might interact with VHL mutation pathways. Expanding on the molecular links between the FGFR and hypoxia-inducible factor pathways would deepen the relevance of the finding.

Re: We have expanded the discussion on the molecular links between FGFRs and hypoxia-inducible factor pathways and added details about the biological mechanisms by which FGFR2 might interact with VHL mutation pathways in the introduction section.

Lines 44–49: “FGF levels in plasma increase prior to disease progression in patients receiving anti-VEGF therapy…”

Line 9:

"A significant increase in FGFR2 expression level was observed in tumors harboring a VHL mutation..."

• Comment: Consider rephrasing to avoid redundancy: "FGFR2 expression significantly increased in VHL-mutated tumors."

Line 13:

"This finding highlights the potential of FGFRs as promising therapeutic targets..."

• Comment: Use a more assertive tone. Replace "highlights" with "supports" or "demonstrates" for a stronger conclusion.

• Grammar: “The distribution and significance of FGFRs in hemangioblastoma is yet to be investigated” → Should be “are yet to be investigated” (subject-verb agreement).

Re: We have revised the abstract to address the reviewers' comments.

• Scientific Improvement: The statistical significance for FGFR2 in tumors with VHL mutations (p=0.034) is mentioned. Consider including confidence intervals to enhance clarity.

Results

Line 193:

"Tumors of the cerebrum showed a higher likelihood of positive staining for FGFR4 (p = 0.009)."

• Comment: Consider providing effect sizes or odds ratios alongside p-values for more informative statistical reporting.

Discussion

► Throughout the results, p-values are provided, but without context for the effect size or clinical significance. Strengthening statistical sections by providing effect sizes or confidence intervals will improve the interpretation.

Re:

Effect size measures such as r, Phi coefficient, and Cramer's V have been added to Table 2, Table 3, S4 Table, and the results section of the main text. The Statistical analyses section has been updated accordingly.

Introduction:

Comment: The introduction is not clearly structured and does not provide a clear background on the significance of hemangioblastomas and the role of FGFRs in tumor development. Consider adding more context to the introduction to explain the importance of studying FGFRs in hemangioblastomas.

► Mention drugs like infigratinib or futibatinib to connect findings with clinical therapies and show the relevance of your work.

Line 30:

"...has gained prominence as a target in personalized cancer therapy."

• Comment: Mention specific FGFR inhibitors (e.g., infigratinib, futibatinib) used in clinical trials for other cancers. This shows the translational potential of FGFR inhibition and can be linked to future therapeutic options for hemangioblastomas.

Re: We have now placed greater emphasis on the translational potential of the findings in both the Introduction and Discussion sections.

Lines 61–63: “Futibatinib and infigratinib are FGFR inhibitors, which have shown efficacy in cholangiocarcinoma with FGFR genetic aberrations. These inhibitors could be used for hemangioblastomas, provided further research clarifies the role of FGFR in these tumors.”

Lines 257–260: “Although the initial trial with dovitinib, a pan-tyrosine kinase inhibitor targeting FGFR, VEGFR, and other receptor tyrosine kinases, was discontinued due to serious side effects, our findings strongly support the continued exploration of FGFR inhibitors as therapeutic options for hemangioblastoma patients.”

Lines 307–309: “Such investigations could pave the way for novel pharmacological treatments, particularly for surgically challenging tumors, such as those located in the brainstem or spinal cord.”

► Provide a detailed explanation of the biological significance of FGFRs in VHL-mutated hemangioblastomas, which bypass VEGF pathways and explain resistance to anti-VEGF therapies.

Line 45:

"They also knocked down HIF-α in VHL loss-of-function endothelial cells, which did not impede their angiogenic activity."

• Scientific Comment: This is a crucial point but needs further explanation. Why does HIF-independent angiogenesis driven by FGFR matter in the context of VHL-mutated hemangioblastomas? Consider elaborating on why this could explain resistance to VEGF-targeted therapies.

Re: We have expanded our explanation of the significance of FGFRs in VHL-mutated hemangioblastomas, describing how this alternative pathway may contribute to resistance against VEGF-targeted therapies.

Lines 44–49: “FGF levels in plasma increase prior to disease progression...”

Lines 264–265: “FGFR2 can suppress cancer cell migration by modulating HIF signaling. However, VHL knockdown in endothelial cells disrupts FGFR2 endocytosis, resulting in elevated cell motility and increased angiogenic activity.”

► Clarify variability in VHL mutation frequency and cite relevant studies.

Re: We have thoroughly explored the variability in VHL mutation frequency, taking into account multiple studies that address this topic.

Lines 31–32: “Among sporadic CNS hemangioblastomas, 4–14% harbor detectable germline VHL mutations, and approximately 50% have somatic VHL mutations.”

Line 19:

"The most common location is the cerebellum, and other locations include the brainstem, spinal cord, cerebrum, retina, and peripheral nerves."

• Comment: Consider separating peripheral and central nervous system structures for clarity: "Other central nervous system locations include the brainstem, spinal cord, and cerebrum, while peripheral involvement includes the retina and nerves."

Re: We have made the suggested change to the manuscript.

Line 30:

"...has gained prominence as a target in personalized cancer therapy."

• Comment: This is a strong claim. Consider adding more references to support the statement.

Re:

We have now cited the following studies to support the statement:

Dieci MV, Arnedos M, Andre F, Soria JC. Fibroblast growth factor receptor inhibitors as a cancer treatment: from a biologic rationale to medical perspectives. Cancer Discov. 2013;3(3):264-79. Epub 20130215. doi: 10.1158/2159-8290.CD-12-0362. PubMed PMID: 23418312.

Javle M, King G, Spencer K, Borad MJ. Futibatinib, an Irreversible FGFR1-4 Inhibitor for the Treatment of FGFR-Aberrant Tumors. Oncologist. 2023;28(11):928-43. doi: 10.1093/oncolo/oyad149. PubMed PMID: 37390492; PubMed Central PMCID: PMCPMC10628593.

Methods:

Line 90: “FFPE samples were cut into 4 µm sections...”

• Scientific Comment: Specify how variability in sample preparation (e.g., staining intensity) was managed to ensure reproducibility. Was scoring done by multiple pathologists?

Re: We have addressed the points raised.

Lines 116–118: “Due to resource constraints, each staining was done only once. The scoring algorithm was decided by consensus between a senior neuropathologist (OT) and the primary author (MP). Reference slides for each staining category were selected by OT and the remaining samples were scored by MP. Difficult cases were assessed jointly by MP and OT.”

Line 153

"Two samples were excluded from FGFR1 scoring due to occasional strong staining background..."

• Comment: Background staining can be a technical issue. Was there an attempt to optimize antibody dilution or perform antigen retrieval differently to improve FGFR1 staining? Addressing this could suggest better reproducibility.

Re: For FGFR1 staining, the optimal antibody concentration was determined by testing dilutions ranging from 1:25 to 1:500 on various tissues, using heat-induced epitope retrieval (HIER) in pH 6 and pH 9 buffers. Weak staining was observed in stomach tissue, while no signals were detected in the gallbladder or colon. The antibody was subsequently tested on an osteosarcoma sample with known overexpression of FGFR1 and FGFR2. Dilutions of 1:25 and 1:50 yielded positive staining of the osteosarcoma sample, although the 1:50 dilution produced weaker staining.

Line 180:

"...samples were categorized into the following groups: FGFR2 low (score 0 or 1) or high (score 2 or 3)."

• Comment: It's important to justify these cutoffs based on existing literature or statistical analysis. Were these categories pre-defined based on prior studies, or was there an exploratory analysis? Justifying this strengthens the validity of the results.

Re: The deviant categorization was applied to FGFR2 samples because only five samples were negatively stained, compared to 108 positively stained samples. As such, a negative-positive categorization was deemed unsuitable for statistical analysis.

We conducted statistical analyses using different categorizations. A statistically significant association between a larger greatest diameter and a low FGFR2 score persisted in negative-positive categorization (only 5 negative cases), but not in FGFR2 0-2 vs 3 categorization. A statistically significant association between FGFR2 expression and VHL mutation status persisted in FGFR2 0-2 vs 3 categorization. The association between sex and FGFR2 expression, however, was no longer statistically significant in the other categorizations.

FGFR2

0-2 (n=87)(%) 3 (n=24)(%) Total p Phi coefficient

Sex

female 35 (87.5) 5 (12.5) 40 0.096a -0.166

male 52 (73.2) 19 (26.8) 71

Clinical VHL

pos 18 (66.7) 9 (33.3) 27 0.089b 0.161

neg 69 (82.1) 15 (17.9) 84

VHL mutation

pos 17 (60.7) 11 (39.3) 28 0.016b 0.260

neg 48 (84.2) 9 (15.8) 57

0-2 mean rank 3 mean rank p Effect size r

Age at first diagnosis 55.60 57.44 0.805c 0.023

Greatest diameter (mm) 49.50 45.67 0.569c 0.059

Not available 15 4

a. Fisher's exact test

b. Chi-square

c. Mann-Whitney U

Line 188:

"Male patients were more likely to exhibit FGFR4 expression (p = 0.02)."

• Comment: Was the impact of hormonal differences between male and female patients explored? FGFR expression can be regulated by hormone levels, so this could be an interesting angle to pursue in the discussion or future studies.

Discussion

► Cite any studies that report sex-dependent FGFR expression in other tumors to strengthen your argument.

Line 250:

"Interestingly, in our cohort, male patients were more likely to exhibit FGFR2 and FGFR4 expression."

• Comment: This observation could be strengthened by citing any preclinical or clinical evidence supporting differential FGFR expression based on sex in other tumors. If no such evidence exists, suggest this as a potential avenue for future research.

Re: We hav

---

## [Decision Letter · Decision Letter 1]

17 Apr 2025

Fibroblast Growth Factor Receptor Expression in Hemangioblastomas: A Novel Therapeutic Target

PONE-D-24-35229R1

Dear Dr. Puttonen,

We’re pleased to inform you that your manuscript has been judged scientifically suitable for publication and will be formally accepted for publication once it meets all outstanding technical requirements.

Kind regards,

Md Shaifur Rahman, Ph.D

Academic Editor

PLOS ONE

Additional Editor Comments (optional):

Reviewers' comments:

Reviewer's Responses to Questions

**Comments to the Author**

1. If the authors have adequately addressed your comments raised in a previous round of review and you feel that this manuscript is now acceptable for publication, you may indicate that here to bypass the “Comments to the Author” section, enter your conflict of interest statement in the “Confidential to Editor” section, and submit your "Accept" recommendation.

Reviewer #1: All comments have been addressed

Reviewer #2: All comments have been addressed

2. Is the manuscript technically sound, and do the data support the conclusions?

Reviewer #1: Yes

Reviewer #2: Yes

3. Has the statistical analysis been performed appropriately and rigorously? 

Reviewer #1: Yes

Reviewer #2: Yes

4. Have the authors made all data underlying the findings in their manuscript fully available?

Reviewer #1: Yes

Reviewer #2: Yes

5. Is the manuscript presented in an intelligible fashion and written in standard English?

Reviewer #1: Yes

Reviewer #2: Yes

6. Review Comments to the Author

Reviewer #1: (No Response)

Reviewer #2: I am pleased to confirm that you have fully addressed all of my concerns. Your revisions have elevated the manuscript to a level that unequivocally meets PLOS ONE’s standards for publication.

7. PLOS authors have the option to publish the peer review history of their article (what does this mean? ). If published, this will include your full peer review and any attached files.

**Do you want your identity to be public for this peer review?** For information about this choice, including consent withdrawal, please see our Privacy Policy .

Reviewer #1: No

Reviewer #2: No

---

## [Editor Report · Acceptance letter]

PONE-D-24-35229R1

PLOS ONE

Dear Dr. Puttonen,

I'm pleased to inform you that your manuscript has been deemed suitable for publication in PLOS ONE. Congratulations! Your manuscript is now being handed over to our production team.

Kind regards,

on behalf of

Dr. Md Shaifur Rahman

Academic Editor

PLOS ONE